# Ecology and Diversity of Weed Communities in the Northern Andes under Different Anthropogenic Pressures

Yessica P. Duque [1], Carlos E. Giraldo-Sánchez [2,*], Mario A. Quijano-Abril [3] and Jose M. Rojas [3,4]

1    Facultad de Ciencias Agropecuarias, Universidad Católica de Oriente, Rionegro 054048, Colombia; yessica.duque8329@uco.net.co
2    Grupo de Investigación de Sanidad Vegetal, Universidad Católica de Oriente, Rionegro 054048, Colombia
3    Grupo de Investigación Estudios Florísticos, Herbario Universidad Católica de Oriente, Rionegro 054048, Colombia; maquijano@uco.edu.co (M.A.Q.-A.); miguel.rojas4@udea.edu.co (J.M.R.)
4    Instituto de Biología, Facultad de Ciencias Exactas y Naturales, Universidad de Antioquia, Medellín 050001, Colombia
\*    Correspondence: cegiral0@gmail.com or csanchez@uco.edu.co

**Abstract:** Weeds can have both positive and negative effects on agricultural environments. However, despite the growing interest in the ecology of weed communities in agricultural areas, a few studies have been carried out in the northern region of the Andes of Colombia, where urban and agricultural expansion have generated highly disturbed scenarios. The aim of this study was to analyze the diversity of vegetation and weed seed banks in three agricultural production systems and a forest ecosystem in the northern Andes of Colombia. Hill numbers were used to compare diversity, Beta diversity to assess changes in composition, and range—abundance—dominance curves at different sites. Likewise, indicator species were analyzed to find species associations to each system. The results revealed differences in the composition of weeds between the forest ecosystem and the agricultural production systems, with higher equitability in the forest ecosystem and higher dominance in agricultural systems. Significant differentiation was observed among the dominant species within each agricultural system, particularly highlighting those species considered pests due to their unique life history traits. These traits confer them with a greater advantage in the face of various anthropogenic selection pressures. These findings highlight the impact of anthropogenic disturbances on the ecological dynamics of weed communities in different ecosystems, which should be considered when planning integrated weed management techniques.

**Keywords:** soil seed banks; surface vegetation; composition; dominance; weeds; forest ecosystems; agricultural production systems

## 1. Introduction

Weeds are adventitious plants that grow in crops without being sown intentionally but can play a crucial role in these systems [1]. These plants have short life cycles, produce abundant seeds, and form seed banks in the soil that ensure their persistence over time in various ecosystems [2,3]. Due to their nature, they can positively or negatively affect agricultural environments, either through soil conservation [4], associated beneficial fauna and allelopathic effects on crops [5], or due to the intense competition exerted by some species for resources such as nutrients, water, and sunlight [6,7]. Some studies suggest a high correlation between severe soil disturbances, with an increase in annual weed communities [8,9]. However, although tillage stimulates the germination of weed seeds dormant in the soil, the effect caused by this and other cultural practices on seed banks depends on the weed species and the interactions with the environment [10–12]. Thus, the diversity of weed species that comprise the seed banks could fluctuate over time and with different crop rotation systems [13]. In this way, understanding the dynamics of weed

communities subjected to different anthropic pressures in different agricultural systems could help design more efficient and environmentally friendly management strategies.

Biodiversity and ecological dynamics of weed communities in agricultural production systems have been the subject of recent studies due to the negative impact that human activities have on these highly modified environments [12]. Authors have suggested that the composition of weed communities in agriculture shows a wide variation related to the different types of crops and the historical uses of the soil [2]. For example, in a study that evaluated seed bank dynamics for five years under a corn–soybean rotation system, seed density in the soil decreased by almost 90% during the first year under a productive system based on corn, and the trend was maintained by rotating with soybeans during the following years [2]. Thus, understanding the diversity variability of weed communities in areas under different anthropogenic pressures, including various crops and associated tasks, is essential to identify problematic species in agricultural production systems and developing more efficient and sustainable management strategies [14]. However, even though in recent years there has been a growing interest in understanding the ecology and diversity of weed communities in agricultural areas with high anthropogenic pressure, little has been studied in the northern region of the Andes, where urban and agricultural expansion have generated highly disturbed scenarios [15].

In the northern part of the Colombian Andes, agriculture, livestock, and urbanization have historically caused significant impacts on plant cover, generating drastic changes in soils and promoting the colonization of weeds, many of them invasive [15,16]. For example, a study carried out in a production system of roses under greenhouse conditions in the Sabana de Bogotá area to evaluate the diversity of weed species in cultivated fields registered 46 species, of which 2, *Cardamine hirsuta* and *Pennisetum clandestinum*, showed a marked dominance with 67% of the total plant cover [17]. Similarly, in a peach orchard in the same region, a low species diversity was found, with a high dominance of *Oxalis corniculata* reaching a 68% coverage [18]. However, some weed species in this Andean area have become a serious problem in forested areas without agricultural pressure, as is the case of the invasive species *Thunbergia alata*, in which an average density of 493 seeds/m$^2$ has been documented, with a viability of 100% stored in the soil [19]. This suggests that the various aspects of the life histories of weeds could be related to the structuring of communities according to anthropogenic dynamics that alter natural ecosystems, both in cover and in seed banks in the soil.

Understanding the diversity of species that comprise seed banks can provide valuable information on the relationships of weeds with their environment [20] and the effect of transforming natural areas into different agricultural contexts. Such information could help minimize the use of herbicides and agricultural inputs, contributing to the conservation and health of ecosystems [21]. In this sense, the aim of this study was to analyze the diversity of vegetation and weed seed banks in four areas of the northern Andes of Colombia under different anthropogenic pressures (little intervened forest, and vegetable, avocado, and livestock farming), with the hypothesis that species diversity changes according to the type of intervention, and that some dominant species can be identified as specific indicators of each system. Consistent with this, the following questions are answered: (a) Can differences in the diversity (Alpha) of the weed communities of the seed banks between the different agricultural production systems and the forest ecosystem be observed? (b) Are there differences in weed species composition in the soil seed banks between the agricultural production systems and the forest ecosystem (Beta diversity)? (c) Which weed species in the seed banks of the soil manage to manifest themselves in its surface vegetation and can potentially be considered "weeds"? (d) Is a higher dominance of weed species observed in the ecosystems most affected by human activity? (e) Which weed species could be considered indicators of each evaluated system?

This work provides novel information related to the composition of the weed communities in the seed banks of little-studied areas with different anthropic impacts. It also provides basic knowledge on patterns of diversity and ecology of different weed species

associated with anthropized ecosystems in one of the most diverse regions in the world and with high rates of endemism, as is the northern area of the Andean mountain range in Colombia [22]. These findings may contribute significantly to integrated weed management plans, prioritizing those that show a marked dominance and that could be considered specific to each productive system evaluated.

## 2. Materials and Methods

### 2.1. Study Area

The study was carried out in the highlands of the Oriente antioqueño area located in the central mountain range (Cordillera) of the Andes, to the southeast of the department of Antioquia, Colombia. In this region, the low montane very humid forest life zone predominates [23] with average temperatures between 18 °C and 21 °C, elevations from 2100 to 2400 m a.s.l. and annual rainfall between 1500 and 4000 mm. This is one of the areas that shows the highest agricultural suitability in the department, being considered a substantial agricultural breadbasket [24], where different production systems have been established, including vegetables, flowers, fruit trees, coffee, sugarcane, and livestock, among others [25].

The sampling was carried out in four sites, three of them corresponding to agricultural production systems farming vegetables (FV), avocado (A), and livestock (L), and a forest fragment identified as a low secondary vegetation cover (VC) [26] (Figure 1).

The forest ecosystem (VC) is in the initial stages of a secondary succession since it has undergone various anthropic intervention phenomena due to deforestation and urban expansion. Its vegetation type is mainly shrubby and herbaceous, with an irregular canopy and occasional trees and vines, with heights of less than five meters [26]. It is dominated by some tree species such as *Cecropia peltata*, *Andesanthus lepidotus*, *Cavendishia pubescens*, *Vismia baccifera*, and *Myrsine guianensis*, and shows an area of 7000 m$^2$. The livestock productive system (L) corresponds to extensive livestock farming for milk production, an area dominated by the Kikuyu grass *Pennisetum clandestinum* used for animal feed. The avocado production system (A) is dedicated to the production of export-type avocados, with an extension of 200,000 m$^2$. Finally, the vegetable production system (FV) includes open fields dedicated to cultivating lettuce, potato, and tomato in an area of 10,000 m$^2$. For weed management in the production systems, the non-selective agrochemical Paraquat, which contains the chemical molecule 1,1′-dimethyl-4,4′-bipyridinium ion dichloride, applied every two months, is used in the livestock area (L) and avocado crop (A). For vegetable crops (FV), selective herbicides with the active ingredient Oxyfluorfen, with the chemical molecule 2-chloro-1-(3-ethoxy-4-nitrophenoxy)-4-(trifluoromethyl)benzene, are used with a frequency of application of every two months.

### 2.2. Sampling Design

A specific sampling was carried out for each production system, including registering soil seed banks and their surface vegetation. In VC and L, the samples were collected haphazardly in a zigzag pattern; in the A, the planted tree area was sampled, and in the FV, samples were collected on the production furrows in areas where weeds were present. In each productive system, an area of 5000 m$^2$ was delimited, establishing five plots of 0.25 m$^2$. Each plot was subdivided into 25 quadrats of 0.1 m. Twenty soil subsamples were extracted from each plot collected with the help of an auger at a depth of 10 cm according to the recommendations of Buhler et al. [2]. The subsamples were grouped, homogenized, and stored in hermetically sealed plastic bags for later analysis, following the methodology described in Bigwood and Inouye [27]. The number of species found in the surface vegetation was recorded in each plot, including the presence or absence of the species without considering their abundances.

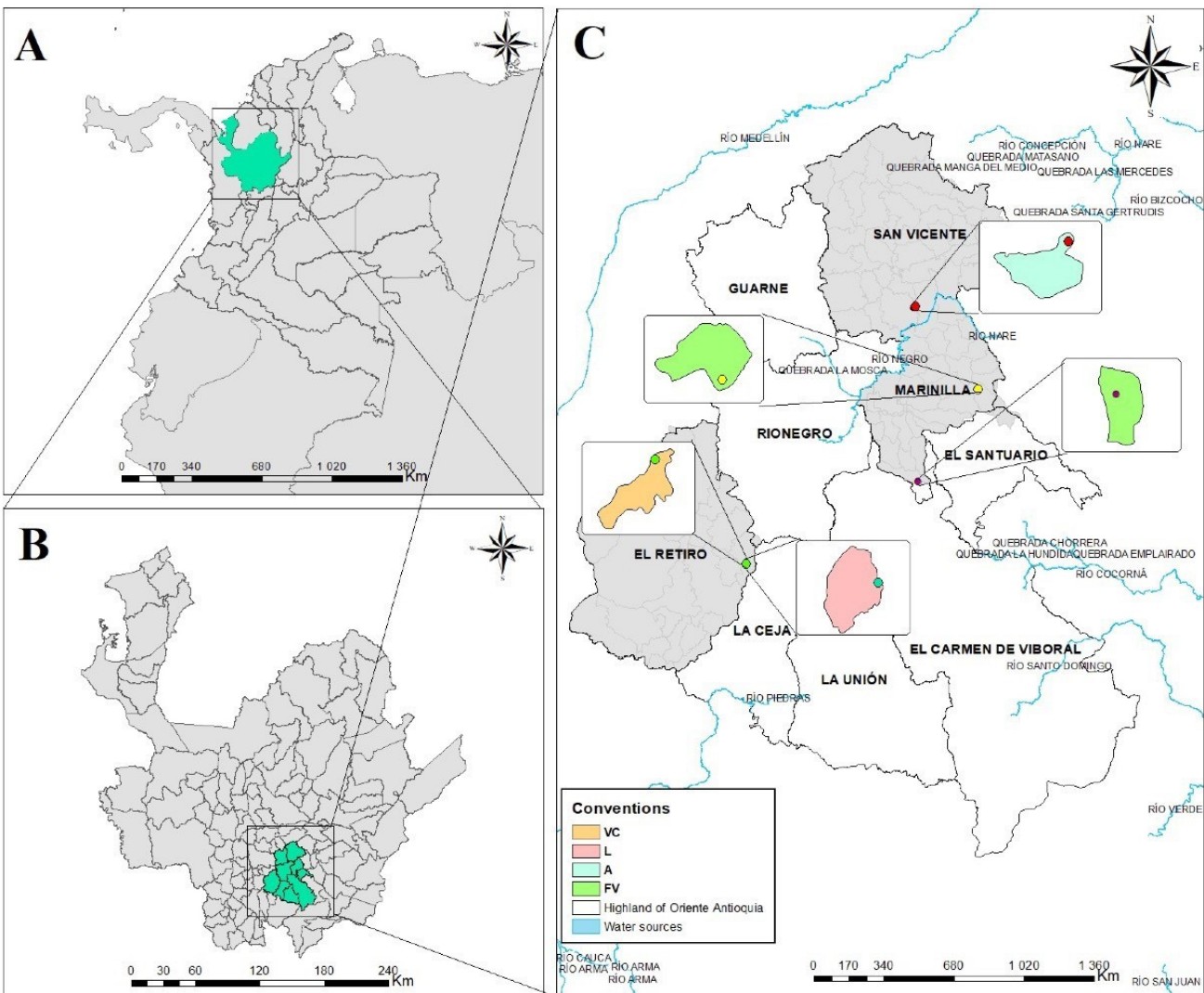

**Figure 1.** Study area. (**A**) Geographical location of the northern Andes, Colombia. (**B**) Location of the department of Antioquia, indicating the highlands of the Oriente antioqueño area. (**C**) Spatial location of the agricultural production systems sampled. VC = low secondary vegetation area, L = livestock production system area, A = avocado production system area, and FV = vegetable production system area.

The seedling emergence method [28] was used to determine the species that comprise the soil seed banks in each production system. In this method, the homogenized material of the soil samples is arranged in 50 × 30 cm germination trays covered with plastic lids to prevent the interference of other weed seeds that may be present in the environment. The trays were left for four months while the seeds germinated. During this time, they were supplied with surface irrigation every 15 days until reaching field capacity, according to the requirements of each tray. All the experiments were carried out under greenhouse conditions with a mean temperature of 26.39 ± 4.85 °C and a relative humidity of 60.83 ± 10.79%. Every 15 days during the four months, the emerged seedlings were counted and recorded, and their taxonomic identification was carried out through comparisons between specimens stored in the reference collection of the herbarium of Universidad Católica de Oriente (HUCO). In addition, taxonomic identification guides from the list of vascular plants of Antioquia [29], the illustrated guide to identifying weed plants of the Marengo agricultural center (CAM) [30], and the guide of frequent weeds in coffee cultivation in Colombia [31] were used.

*2.3. Data Analysis*

Only zero-order (q0) Alpha diversity was calculated for the surface vegetation in the four productive systems assessed and used to compare the weed species found in the soil seed bank with their surface vegetation. Likewise, the Alpha diversity of the soil seed banks of each productive system was computed using the numbers of species equivalents [32,33]. For Order 0 diversity or richness (q0), the number of species found in the soil seed banks was considered excluding abundance. Order 1 diversity, calculated as the exponential of the entropy of the Shannon index (q1), analyzes the diversity weighted by the relative frequency of each species in the community. Finally, Order 2 diversity calculated as the inverse of the Gini–Simpson index (q2) considers the most abundant or dominant species in the community, discarding rare species. From the data obtained, the confidence intervals were calculated [34]. At the same time, the completeness of the seed banks sampling in each productive system was evaluated using the sample coverage (*Cm*) proposed by Chao and Jost [34] and based on the following equations:

$$
\begin{aligned}
E(Cm) &= \sum_{i=1}^{S} pi[1 - (1 - pi)^m] \\
&= 1 - \sum_{i=1}^{S} pi(1 - pi)^m, m > 0,
\end{aligned}
\tag{1}
$$

where

*Cm* = sampling coverage
*S* = total number of species sampled
*pi* = relative abundance of the *i*th species
*m* = sample size.

This value indicates the proportion of the total community represented by the weed species found in each study area. When its value approaches 100%, the complete sample is considered concerning the sampling effort made [34]. The estimation was performed in the iNEXT web application [34]. In addition, rank–abundance curves were established for each production system to compare the dominance of the weed species in the soil seed banks [35].

The Beta diversity was calculated for three orders (β0, β1, and β2) following the multiplicative partition of diversity [36] to evaluate the change in the composition of weed species, seed banks, and between each production system. The Baselga partitioned Beta index was also calculated [37] to estimate the proportion of turnover and nesting in differentiating communities between sites. Additionally, comparisons were made between the composition of the species found in the soil seed bank with the superficial vegetation in each of the productive systems using Whittaker's complementarity index [38], which exclusively contemplates matrices of presence and absence of the species between the compared units. The above is performed to estimate the representation of the seed bank in the coverage of each site. Finally, a species indicator analysis was performed to estimate the weed species best associated with each production system using the indicator value method (*IndVal*) according to Dufrene and Legendre [39] and calculated as follows:

$$
\textbf{\textit{IndVal}} = specificity \times Fidelity \times 100,
\tag{2}
$$

where

*Specificity = Nindij/Nindi*
*Fidelity = Ntrapij/Ntrapj*
*Nindij* = the average number of individuals of species *i* in type *j* habitat
*Nindi* = the sum of the average number of individuals of species *i* in all habitat types
*Ntrapij* = the number of individuals in habitat *j* where species *i* is present
*Ntrapj* = the total number of individuals in habitat *j*.

The species with high IndVal (higher than 50%) were considered the best indicators of the productive systems, while the species with low percentages (less than 25%) were not considered indicators [39]. Statistical significances (*p*-values) were estimated using 9999 random permutations of sites between groups. The data were analyzed in the statistical software Past version 4.13 [40].

## 3. Results

### 3.1. Species Richness

From the soil seed banks, a total of 3332 weed seedlings were recorded emerging from the four productive systems, belonging to 38 species, 36 genera, and 21 botanical families. Asteraceae was the most representative family with 12 species (31.57%) of the total number of emerged seedlings, followed by Cyperaceae and Polygonaceae with three species each (7.89%), and Caryophyllaceae with two species (5.26%). The other families were only represented by a single species (Table 1). The highest seedling emergence in the soil seed banks was constituted by genera *Cardamine* (723), *Trixella* (636), *Verbena* (380), *Polygonum* (375), *Oxalis* (200), *Cyperus* (277), and *Gnaphalium* (179). The productive system with the highest number of individuals was L with 1872, followed, with a much lower number, by A with 534, FV with 525, and VC with 401, i.e., the lowest number of individuals. The species predominating in the soil seed banks of the four productive systems evaluated were *Cardamine hirsuta* with 282 seedlings and *Polygonum nepalense* with 136 individuals in FV. Conversely, *Trixella arvensis*, *Oxalis corniculata*, and *Cyperus odoratus* dominated in L with a total number of 605, 152, and 83 seedlings, respectively (Table 1).

**Table 1.** Weed species abundance found in the soil seed banks of four productive systems evaluated in the Oriente antioqueño region, northern Andes, Colombia.

| Family | Species | VC | L | A | FV |
|---|---|---|---|---|---|
| Amaranthaceae | *Amaranthus viridis* L. | | | | 3 * |
| Apiaceae | *Centella asiatica* (L.) Urb | 7 * | 6 | | |
| Araliaceae | *Hydrocotyle umbellata* L. | 6 * | | | |
| Asteraceae | *Ageratum conyzoides* L. | 20 * | | 12 * | |
| | *Erechtites valerianaefolia* C.E.C. Fisch | 5 * | 22 * | 1 * | |
| | *Porcellites radicata* (L.) Cass. | 15 * | 7 * | 48 * | |
| | *Jaegeria hirta* (Lag.) Less. | 3 * | 6 * | 5 * | |
| | *Gnaphalium americanum* Mill. | | 110 * | 60 * | 9 * |
| | *Sonchus oleraceus* L. | | 18 * | | |
| | *Artemisia vulgaris* L. | | 5 | | 20 * |
| | *Conyza bonariensis* (L.) Cronquist | | 1 * | 9 * | |
| | *Emilia sonchifolia* (L.) DC. | | | 6 * | |
| | *Galinsoga quadriradiata* Ruiz and Pav. | | | | 13 * |
| | *Senecio vulgaris* L. | | | | 3 * |
| | *Acmella oppositifolia* (Lam.) R.K. Jansen | | | | 1 * |
| Brassicaceae | *Cardamine hirsuta* L. | 2 | 263 | 176 | 282 |
| Caryophyllaceae | *Stellaria media* (L.) Vill. | 46 | 1 | 8 * | 2 * |
| | *Drymaria villosa* Schltdl. and Cham | | | 15 * | |
| Commelinaceae | *Commelina diffusa* Burm. f. | 12 * | 2 * | | 1 * |
| Convolvulaceae | *Ipomoea purpurea* (L.) Roth | | | | 1 * |

**Table 1.** *Cont.*

| Family | Species | VC | L | A | FV |
|--------|---------|----|----|----|----|
| Cyperaceae | *Cyperus odoratus* L. | 67 | 83 | 48 * | 12 * |
| | *Cyperus rotundus* L. | 51 | 2 | 14 * | |
| | *Kyllinga erecta* Schumach. | 42 | 3 | | |
| Fabaceae | *Mimosa albida* Humb. and Bonpl. ex Willd | 3 * | | | |
| Melastomataceae | *Chaetogastra kingii* (Wurdack) P.J.F. Guim. and Michelang. | 14 * | | | 3 * |
| Lythraceae | *Cuphea carthagenensis* (Jacq.) J.F. Macbr. | 20 * | 4 * | | |
| Iridaceae | *Sisyrinchium micranthum* Cav. | 27 * | 10 * | 4 * | |
| Lamiaceae | *Trixella arvensis* (L.) Fourr. | 1 | 605 * | 4 * | 26 * |
| Phyllanthaceae | *Phyllanthus niruri* L. | 2 * | | | |
| Rubiaceae | *Richardia scabra* L. | 32 * | 5 | 20 * | |
| Oxalidaceae | *Oxalis corniculata* L. | 1 | 152 * | 39 * | 8 * |
| Poaceae | *Bromus* sp. L. | 16 * | | | |
| | *Paspalum paniculatum* L. | | 11 * | 50 * | 5 * |
| Polygonaceae | *Polygonum nepalense* Meisn. | 4 * | 125 * | 9 * | 136 * |
| | *Polygonum segetum* Kunth | | 1 * | | |
| | *Rumex crispus* L. | | 49 * | | |
| Plantaginaceae | *Plantago major* L. | 5 * | 1 * | 6 * | |
| Verbenaceae | *Verbena litoralis* Kunth | | 380 * | | |
| N° individual per site | | 401 | 1872 | 534 | 525 |
| N° species per site | | 23 | 25 | 19 | 16 |

VC = low secondary vegetation area, L = livestock production system area, A = avocado production system area, and FV = vegetables production system area. * Species registered both in the seed bank and in the surface vegetation.

The plots of the four-soil seed bank production systems exhibited variability in terms of both richness and abundance. Regarding richness, the weed communities displayed no significant differences in the number of species. However, when it came to abundance, notable variations were observed in relation to the number of individuals present in each of the production system plots (Table 2).

**Table 2.** Average richness and abundance with standard deviations (in brackets) across sampled plots in four production systems of the Oriente antioqueño region, Northern Andes, Colombia.

| | L | VC | A | FV |
|--|---|-----|---|-----|
| Richness | 12.8 (±3.49) | 12.2 (±5.67) | 8.2 (±2.16) | 7.2 (±2.77) |
| Abundance | 374.4 (±148.79) | 80.2 (±56.82) | 106.8 (±41.49) | 105 (±63.65) |

The total diversity observed in the soil seed banks of the four productive systems was 38 species, with a sampling coverage of 0.999, indicating a representative sampling. The richness and diversity of the seed banks differed between the productive systems; the diversity of L showed the highest richness value of the soil seed banks (q0) with 25 species, followed by VC with 23 and A with 19. On the other hand, FV registered the lowest richness value with 16 species (Table 3). Even though L obtained a higher richness (q0), the weed community of the seed banks was more equitable in VC since its (q1) and (q2) diversity values were higher compared to those of the other systems. Furthermore, regarding the diversity weighted by the most abundant species (q2), VC registered a higher value of

11.07 equivalent species once again, followed by A with 6.40, L with 5.47, and FV with 2.77 (Figure 2).

**Table 3.** Alpha diversity values for the weed species registered in four productive systems sampled in the Oriente antioqueño region, northern Andes, Colombia.

| Diversity | | VC | L | A | FV | Total |
|---|---|---|---|---|---|---|
| Observed | q0 | 23 [18.89; 27.11] | 25 [19.16; 30.84] | 19 [17.59; 20.41] | 16 [8.37; 23.63] | 38 [32; 43] |
| | q1 | 14.11 [13.03; 15.19] | 7.75 [7.37; 8.13] | 9.89 [9.10; 10.68] | 4.30 [3.84; 4.76] | 13.40 [12.87; 13.94] |
| | q2 | 11.07 [9.80; 12.35] | 5.47 [5.16; 5.77] | 6.40 [5.48; 7.33] | 2.77 [2.50; 3.04] | 8.58 [8.18; 8.97] |
| Estimated | q0 | 23.86 [18.42; 29.30] | 27.53 [18.54; 36.51] | 19.00 [17.78; 20.22] | 18.19 [8.52; 27.86] | 40.19 [32.80; 47.58] |
| | q1 | 14.39 [13.28;1 5.49] | 7.79 [7.40; 8.18] | 10.01 [9.13; 10.88] | 4.35 [3.88; 4.81] | 13.46 [12.92; 14.00] |
| | q2 | 11.21 [9.89; 12.54] | 5.47 [5.17; 5.78] | 6.44 [5.55; 7.33] | 2.77 [2.50; 3.04] | 8.59 [8.19; 8.98] |
| Sample coverage | | 0.995 | 0.998 | 1 | 0.994 | 0.999 |

VC = low secondary vegetation area, L = livestock production system area, A = avocado production system area, and FV = vegetables production system area. Note: The numbers inside the [ ] indicate the 95% confidence interval.

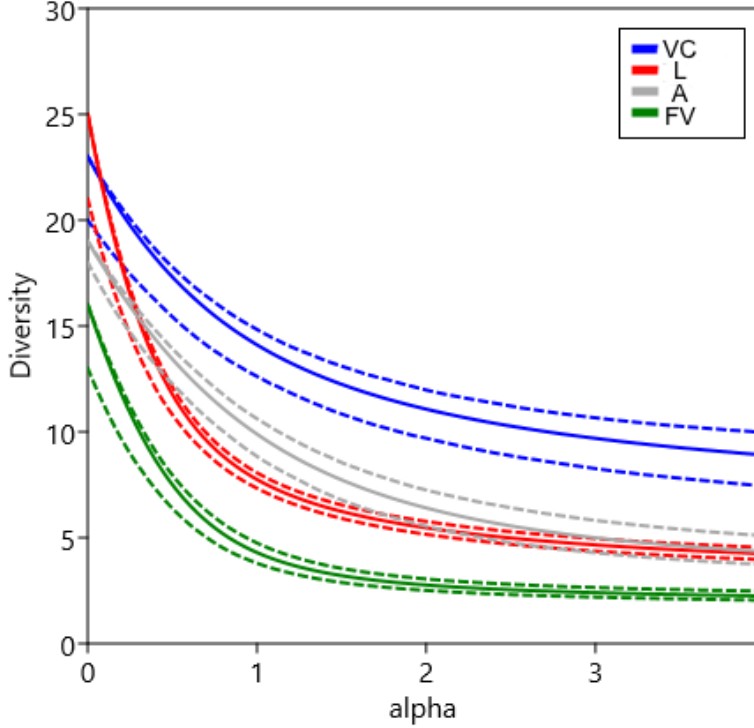

**Figure 2.** Alpha diversity profiles based on the number of equivalent species from the soil seed banks of weeds from the four production systems sampled in the Oriente antioqueño region, northern Andes, Colombia. VC = low secondary vegetation area, L = livestock production system area, A = avocado production system area, and FV = vegetables production system area. Alpha indicates order of diversity q0, q1 and q2. Dashed lines indicate confidence intervals (95%).

The range–abundance graph revealed fluctuations in the dominance of weed species from the seed banks in each production system. In VC, no species showed a clear dominance. On the other hand, in the L productive system, a higher dominance of species was demonstrated, with a total of six weed species with values higher than 100 individuals (*Trixella arvensis* with 605 individuals, *Verbena litoralis* with 380, *Cardamine hirsuta* with 263, *Oxalis corniculata* with 152, *Polygonum nepalense* with 125 and *Gnaphalium americanum* with 110 individuals). In A, a single species, *Cardamine hirsute*, dominated with 176 individuals, while in FV, two dominant species, *Cardamine hirsuta* with 282 and *Polygonum nepalense* with 136 individuals, were found (Figure 3).

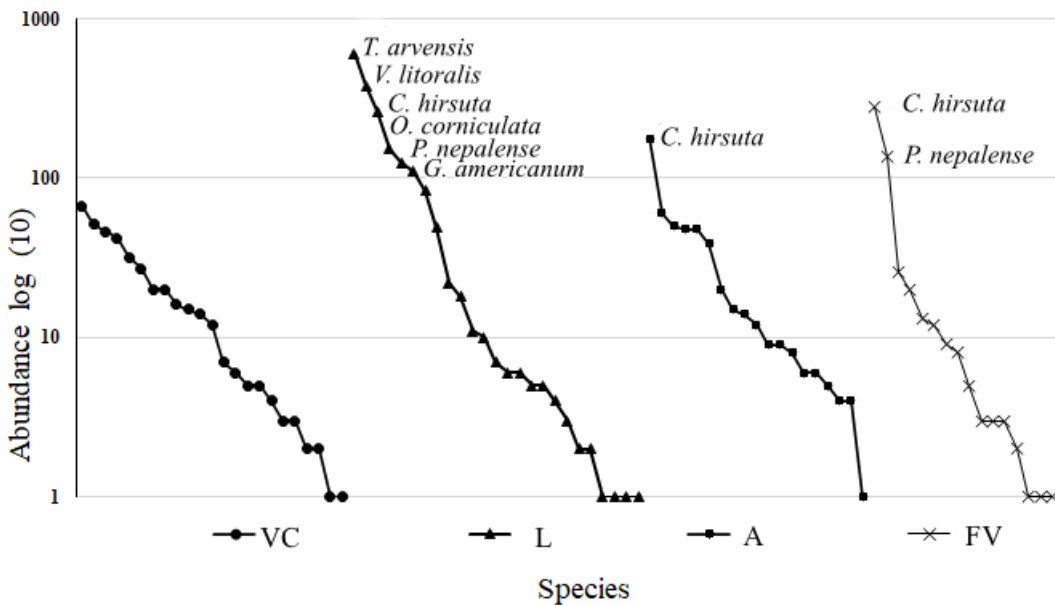

**Figure 3.** Weed species range–abundance curves in the seed banks of the sampled production systems. The names of species with abundances higher than 100 individuals are provided. VC = low secondary vegetation area, L = livestock production system area, A = avocado production system area, and FV = vegetable production system area.

*3.2. Species Composition*

3.2.1. Seed Bank Composition

Richness (β0) between the comparisons of the VC and the L and A productive systems showed a low dissimilarity in the composition of weed species of the seed banks since they showed values of higher than 50% similarity, represented by [0.29; 0.33; 0.27]. In addition, when the VC, L, and A are compared with FV, the richness (β0) tends to increase the dissimilarity of the species. Regarding (β1), the comparison between VC and FV showed a high dissimilarity, represented by a 79% variation between the species of these systems. Regarding (β2), when comparing the forest ecosystem (VC) with the agricultural systems, there was a high dissimilarity of species, registering values of higher than 50% of difference, represented by [0.84; 0.60; 0.91] (Table 4). In addition, the variations in the composition of the weed communities were mainly caused by the high turnover values of the species (βC-bal), reflected in the paired comparison between the forest ecosystem with the agricultural production systems, recording values of [0.690; 0.655; 0.935] (Table 5).

**Table 4.** Beta diversity of the soil seed banks of weeds in multiplicative partition among the four productive systems evaluated in the Oriente antioqueño region, northern Andes, Colombia.

|  | VC × L | VC × A | VC × FV | L × A | L × FV | A × FV |
|---|---|---|---|---|---|---|
| β0 | 0.29 | 0.33 | 0.58 | 0.27 | 0.51 | 0.54 |
| β1 | 0.41 | 0.46 | 0.79 | 0.24 | 0.21 | 0.30 |
| β2 | 0.84 | 0.60 | 0.91 | 0.43 | 0.41 | 0.15 |

VC = low secondary vegetation area, L = livestock production system area, A = avocado production system area, and FV = vegetables production system area.

3.2.2. Composition of the Seed Banks and Their Surface Vegetation

The Beta diversity in the four productive systems showed a high similarity between the composition of weed species that comprise the soil seed banks concerning their surface vegetation, showing values higher than 50%, corresponding to the 76.19% of similarity in VC, 77.27% in L, 93.75% in FV, and 100% in A. Further, the species found in the soil seed banks were also registered in the superficial vegetation of this productive system.

**Table 5.** Beta partitioned diversity of the soil seed banks of weeds among the four productive systems evaluated in the Oriente antioqueño region, northern Andes, Colombia.

|  |  | β.Bray.bal (Turnover) | | | |
|---|---|---|---|---|---|
|  |  | VC | L | A | FV |
| β.Bray.gra (nesting) | VC | 0 | 0.69 | 0.655 | 0.935 |
|  | L | 0.2 | 0 | 0.299 | 0.133 |
|  | A | 0.048 | 0.389 | 0 | 0.571 |
|  | FV | 0.008 | 0.487 | 0.003 | 0 |

VC = low secondary vegetation area, L = livestock area, A = avocado production system area, FV = vegetables production system area.

### 3.2.3. Indicator Species

In the VC, three weed species were registered as indicators of the productive system, with IndVal values higher than 60%. These species were *Cyperus rotundus* (60.9%; *p* = 0.0048) and *Stellaria media* (64.56%; *p* = 0.0053), and the species that had the highest representativeness was *Kyllinga erecta* (74.67%; *p* = 0.0024). L registered the following indicator species: *Oxalis corniculata* (76%; *p* = 0.001), *Verbena litoralis* and *Sonchus oleraceus* (80%; *p* = 0.0021), *Trixella arvensis* (95.13%; *p* = 0.0003) and *Rumex crispus* with 100% (*p* = 0.0003). We recorded *Conyza bonariensis* as an indicator species but with a value lower than 60% (54%; *p* = 0.0186). Lastly, in FV, *Galinsoga quadriradiata* (80%; *p* = 0.0013) was the only indicator species for the system (Figure 4).

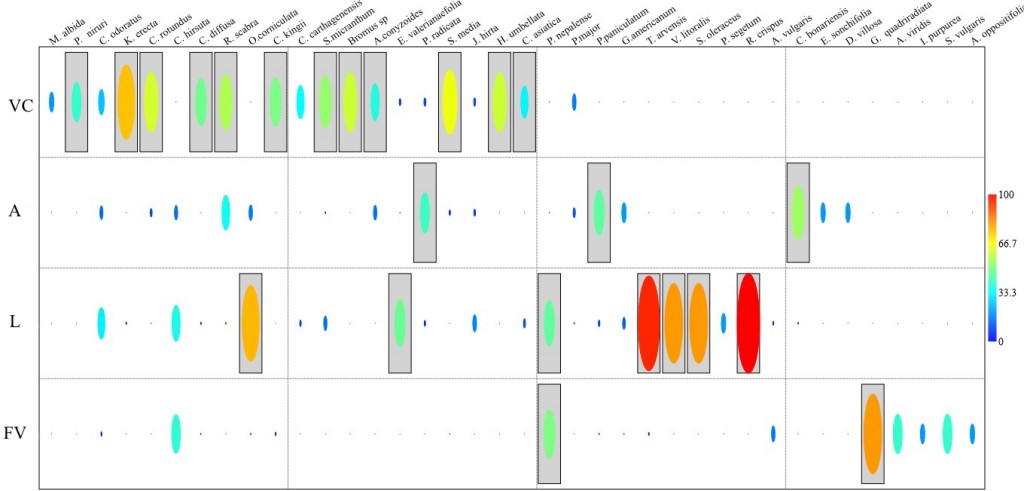

**Figure 4.** Indicator analysis of weed species of the four production systems sampled in the Oriente antioqueño region, northern Andes, Colombia. VC = low secondary vegetation area, A = avocado production system area, L = livestock productive system area, and FV = vegetables productive system area. Yellow, orange, and red colors correspond to InVal values higher than 60%.

## 4. Discussion

Human activities exerted on ecosystems can have an impact on the ecological dynamics of weed communities, as well as on the composition and dominance of species [41,42]. Among these activities, agriculture and deforestation have been identified as the leading causes of impacts on natural areas [42,43]. Although several studies have documented variations in the diversity of weed communities related to disturbances [12,14,43–45], in the Andes, there is a shortage of research that evaluates the effect of anthropogenic disturbance associated with different agricultural practices on weed vegetation cover [17,18]. In this work, the taxonomic diversity of weed communities in one of the areas of greatest anthropogenic pressure, the northern Andes in Colombia, was studied by comparing the surface cover and the soil seed banks in three intensive agricultural production systems and a slightly intervened forest ecosystem.

Our results indicate that, despite finding no significant differences in weed richness between the compared areas (Alpha q0), variations were observed in the composition of the communities, mainly reflected in the high turnover values ($\beta C\text{-}_{bal}$) and minor nesting ($\beta C\text{-}_{gra}$). In addition, the diversity profiles were less equitable in the agricultural production systems compared to the forested area, suggesting a higher dominance of species considered "weeds", which have a negative economic impact on these production systems. Additionally, the second-order Beta diversity ($\beta^2$) for the seed banks [0.15; 0.91] suggests that weed communities under different anthropic pressures have similar structuring patterns in which dominance increases, but differences in dominant species are probably the result of contrasting life histories that allow their response in different ways to selection pressures generated by human activity.

Several studies have documented the differences in the composition and abundance of weed species between production systems and forest ecosystems; they attribute these changes to variations in the growth habits of the species and the agricultural management supplied to the weeds to control their growth [12,46–49]. Likewise, previous studies have shown that the composition of weed communities in soil seed banks is influenced by human activities [2,43]. Similarly, recent research suggests that differences in weed species composition between natural ecosystems and agricultural production system areas are subject to constant anthropogenic disturbances caused by tillage, with higher dominance of weed species in agricultural systems that have experienced major interventions [12,43,49–51]. Thus, the disturbance frequencies governed by tillage in production systems generate notorious changes in the weed community, where highly disturbed environments tend to be simpler and less stable in the abundance of weed populations [12,43,49–51]. For example, as a disturbance in soil seed banks increases in corn, soybean, and oat cropping, the dominance of weed species capable of adapting to these areas tends to increase [52,53]. Similarly, it has been observed that the weed communities in the forest cover do not reflect dominance in the most conserved areas, in contrast to the most intervened areas such as cropping fields since the species of the conserved ecosystems are less abundant and competitive [42].

However, cases have also been reported where, in agricultural production system areas, the abundance, diversity, and uniformity of the weed community in the seed banks tend to increase as the disturbances caused by soil tillage decrease [49,54]. Gurber and Claupein [55] recorded a higher abundance of weed species in more conserved areas compared to those highly disturbed. These authors relate their results to the high capacity of weeds to produce a large number of seeds that persist in the soil forming seed banks, also suggesting that the physical and chemical characteristics of the soil can influence the diversity of weeds in less disturbed ecosystems.

Sharp [56] argued that the differences in the life cycles of the weed species (annual vs. perennial) may influence the dominance of the species. It has been documented that environments with high anthropogenic disturbances favor the growth of weeds with annual cycles, which have the capacity to grow rapidly when tillage is interrupted, reaching reproductive maturity in a short time [8,9,12]. This allows the species produce a large number of seeds in a single season, increasing their ability to disperse and colonize open areas [12,57–59]. In addition, it has been indicated that the notable abundance of weed species in cultivated areas and pastures may be related to reproductive strategies and seed dispersal mechanisms, allowing the species the expansion to new habitats and quick colonization of those disturbed areas [12,42]. These studies support the current results since, in the agricultural production systems, there was a marked dominance and abundance of weed species with annual cycles, including *Trixella arvensis, Verbena litoralis, Cardamine hirsuta, Polygonum nepalense,* and *Gnaphalium Americanum.* The only exception was *Oxalis corniculata.* The species found in the current study as dominant, which could be considered weeds in agricultural production systems, show certain characteristics that allow them the domination of agricultural areas. For example, it has been reported that the high abundance of the species *Cardamine hirsuta* in horticultural crops could be related to its ability to easily

adapt to disturbed environments and grow in open habitats with higher availability of direct light [17]. Likewise, its reproductive capacity offers it advantages to dominate, since it is a species with a short cycle (annual) and a high germination potential; it has been recorded to produce approximately 5000 seeds with germination percentages higher than 90% [60,61]. In addition, this species has self-dispersal mechanisms, favoring its dominance in anthropized environments [17,60,62]. Similarly, it is argued that the dominance of *Oxalis corniculata* in crop fields is related to its polymorphic reproduction since it is a species capable of easily reproducing both by seeds and vegetatively [63,64]. Conversely, *Polygonum nepalense* has been reported as a weed in productive systems of the Colombian Andes, and its dominance is due to its reproductive traits through the production of a large number of seeds (approximately $27,900/m^2$); the seeds have the ability to survive for long periods in the soil and form seed banks, in addition to having a wide range of adaptation to disturbed ecosystems [17,65,66].

Other findings correspond to the fact that the diversity of soil seed banks is higher in the less disturbed ecosystem. These findings are reflected in high values of first- and second-order Alpha diversity (q1 and q2) in the forest ecosystem, presenting a more equitable and homogeneous behavior in terms of its species, and are supported by previous research. For example, Mitja and Miranda [42] found similar results, indicating that forest covers with some degree of conservation may show a higher diversity of weed species, and the diversity in this type of habitat may be related to the stability dynamics of forest ecosystems [49]. In line with these findings, in the current study, the weed community in the soil seed banks and the surface vegetation of the forested area were found not to show competitiveness characteristics. These results support the idea that less intervened ecosystems offer favorable conditions for the coexistence of multiple weed species in an equilibrium [49,67].

On the other hand, the weed communities in the soil seed banks in the agricultural production systems showed a lower first- and second-order Alpha diversity compared to the forest ecosystem. These results agree with previous investigations, indicating that the low diversity of weeds in soil seed banks is a consequence of the high disturbance pressure exerted by man on agricultural systems for their control [49,68], where intensive management practices can favor the growth and establishment of some weed species while restricting the development of others [49,67,68]. Therefore, agricultural management, including selection pressures in production systems, could be a contributing factor to the low weed diversity observed in these environments [12,41,49,51,57,67].

It is crucial to recognize that while the findings imply that the sampling effort for each production system was sufficient and that the observed and expected diversity are in alignment, logistical constraints tied to material collection limited our ability to conduct sampling at only one site within each production system for this study. Despite this constraint, species estimation indicates a marginal increase in just two additional species with doubled sampling efforts. As such, the decision to expand the number of plots at this specific site seems to offer minimal impact. However, an exception arises in the context of the vegetable production system, wherein further sampling could potentially unveil a modest rise in newly identified species. This circumstance suggests that future research endeavors should consider this system as a potential focal point for deeper exploration.

The agricultural production systems assessed have historically experienced various types of agricultural management to control weed populations. Among these management practices, the application of herbicides and overgrazing stand out in livestock areas. In avocado and vegetable production systems, soil tillage practices are highlighted. Some studies have reported that the diversity of weeds in soil seed banks and their surface vegetation is influenced by the type of management used in each production system, suggesting that the use of herbicides, soil tillage, or grazing result in communities of different weeds [8,12,41,43,49,52,67,69]. However, although agricultural management practices in production systems can reduce the diversity of weeds, it has been observed that the abundance of these species does not always decrease and may even tend to increase the

dominance of a few species that have the capacity to easily compete with other weed species [43–45,70], agreeing with the results found in the current study.

The results suggest that understanding the variability of the diversity of weed communities in soil seed banks and their surface vegetation in areas under different anthropogenic pressures in the northern Colombian Andes provides valuable information on the ecological dynamics of weeds species, which could facilitate the identification of those that really represent a problem in agricultural production systems. In addition, a clear understanding of how different agricultural management practices interact to condition weed communities is a key component in the development of integrated weed management programs focused on agricultural efficiency and environmental sustainability based on ecological approaches that promote the biodiversity of ecosystems [42,45,46,54,71–75].

## 5. Conclusions

The composition of the weed communities in soil seed banks and the superficial vegetation differed between the little-intervened forest ecosystem and the agricultural production systems assessed. A greater equitability was observed in the forest ecosystem, while there was a higher dominance of some species in the agricultural systems. Likewise, the highest differentiation occurred in the species composition between sites and the dominant species in the agricultural systems, those with a negative impact on agricultural production standing out probably due to their life history traits that could make them more successful in the face of different anthropic selection pressures in each productive system. Thus, understanding the dynamics of weed communities subjected to various anthropic pressures in different production systems could help design more efficient and environmentally friendly weed management strategies.

**Author Contributions:** Conceptualization, Y.P.D. and C.E.G.-S.; methodology, Y.P.D., C.E.G.-S. and J.M.R.; Software, C.E.G.-S.; validation, C.E.G.-S. and M.A.Q.-A.; formal analysis, C.E.G.-S.; research, Y.P.D.; Resources, data curation, C.E.G.-S.; writing—original draft preparation, Y.P.D. and J.M.R.; writing—proofreading and editing, Y.P.D., C.E.G.-S., M.A.Q.-A. and J.M.R.; display, C.E.G.-S.; supervision, C.E.G.-S.; project management, Y.P.D. and C.E.G.-S.; fund acquisition, M.A.Q.-A. All authors have read and agreed to the published version of the manuscript.

**Funding:** This research was funded by the "Dirección de Investigación Desarrollo e Innovación, Universidad Católica de Oriente (UCO)", Project code: 202207.

**Institutional Review Board Statement:** On behalf of UniversidadCatólica de oriente, ANLA approved the biological collection of plant in the study area (permit number 1726, of 2019).

**Data Availability Statement:** Data are available from the corresponding author upon request.

**Acknowledgments:** We express our gratitude to Wilmar Sánchez for his support in the fieldwork and Daniela Alzate for the technical support in preparing the different maps. We also thank the owners of the properties where the study was carried out and the herbarium of the Universidad Católica de Oriente (HUCO) for their support throughout the entire research process.

**Conflicts of Interest:** The authors declare no conflict of interest.

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
