# Peer review of "Ecology and Diversity of Weed Communities in the Northern Andes under Different Anthropogenic Pressures"

_diversity, doi:10.3390/d15080936_

Round 1

Reviewer 1 Report

Review of “Ecology and diversity of weed communities in the northern Andes under different anthropogenic pressures” by Yessica P. Duque 1, Carlos E. Giraldo 2* Mario A. Quijano3 and Jose M. Rojas3,4

 Summary

This paper examines the diversity of seed banks of four agricultural production systems in the northern Andes.  The aim of the paper is to determine the relationship between the kind of agricultural practice and the diversity of the soil seed bank and the level of similarity of the seed bank to the extant weed vegetation.  The paper is well organized and written.  I found no issues with the grammar.  The tables and figures are clear (for the most part) and appropriate.  The citations are extensive and seem to be appropriate.  The subject of the paper is relevant and important to understanding the relationship between agricultural practices and the associated weed communities.  The analyses of diversity and the dominant weed species seem appropriate.  The discussion of the difference of the “forest” system and the other three seems reasonable.

 General comments

My main concern with the paper is it generalizes about the four different production systems, but it seems that only one site for each community was sampled.  There is great temporal and spatial variation in both the extant vegetation and in the seed bank.  For example, my field converted from dominance by thistle to dominance by goldenrod in a two-year period.  We cannot assess the spatial variation within production systems of this study because no measurement of variation among plots within production systems is given and with only one site for each community, we have no way to know if the sites chosen are representative of that community.  Using one site to represent a type of agricultural community is inadequate.  Twenty trays were used to assess the seed banks.  This study would have been much stronger if five sites had been sampled for each of the four different production systems.  That would have resulted in 100 trays being used which with four authors should not have been overly burdensome.  If the work is not redone to better represent each production system with more sites for each community, the paper should be published as a brief note.

Specific comments

Line 138.  No indication of the type and frequency of use of herbicides is given for the four production systems.  The discussion correctly states that herbicide use is an important determinant of the seedbank composition.

 Line 141.  “Random sampling” has a specific meaning.  It means that each plot has an equal chance of being selected.  It usually involves assigning numbers to each potential plot and using a random number table or generator to select the plots to be sampled.  I doubt that this was done.  “Haphazard sampling” would be more descriptive of what was done.

 Line 151.  How did you account for dormant seeds that were in the seedbank but that did not germinate?  Do seeds in this region have after-ripening requirements?

 Line 156.  How did you account for seedlings that emerged and died before the 4-month end point?  Why were seedlings not assessed more frequently than once at the end of 4 months?  It would have been better if the seedlings had been assessed at least monthly and the soil mildly disturbed after each assessment.

 Line 154.  What were the conditions during the 4-month germination duration?  Specifically, what was the level and variation in light?  What was the average temperature and how did it vary over the course of a day and the 4-month duration?

 Table 1 would be improved by adding sums at the bottom of each column.

 Figure 2 needs more explanation.  What are alpha 0-3?  I did not see that defined in the text.

 Line 318.  No indication is given for variability among the 5 plots for each production system.

Reviewer 2 Report

This is an interesting study of average quality. However, there is a clear lack of novelty. One of the main findings is that there were differences in the composition of weeds between the forest ecosystem and the agricultural production systems. This is something widely known and rather expected. Moreover, to have comparable results, the sampling procedure should be the same. However, in the methods & materials section, authors state that "In VC and L, the samples were collected randomly in a zigzag pattern; in the A, the planted tree area was sampled, and in the FV, samples were collected on the production furrows in areas where weeds were present.". This is something that should be avoided and authors are encourage to follow the same protocol and be specific on the season that they made the samplings and justify their choices. Furthermore, authors should improve the English language and revise their study (indicatively: a few studies (instead of few studies) or avoid "A higher differentiation").

This is an interesting study of average quality. However, there is a clear lack of novelty. One of the main findings is that there were differences in the composition of weeds between the forest ecosystem and the agricultural production systems. This is something widely known and rather expected. Moreover, to have comparable results, the sampling procedure should be the same. However, in the methods & materials section, authors state that "In VC and L, the samples were collected randomly in a zigzag pattern; in the A, the planted tree area was sampled, and in the FV, samples were collected on the production furrows in areas where weeds were present.". This is something that should be avoided and authors are encourage to follow the same protocol and be specific on the season that they made the samplings and justify their choices. Furthermore, authors should improve the English language and revise their study (indicatively: a few studies (instead of few studies) or avoid "A higher differentiation").

Reviewer 3 Report

Your work is well conducted and presented.

Author Response

Thanks for your comments 

Round 2

Reviewer 1 Report

I see that the other reviewers have recommended that the paper be published.  I would recommend publishing the paper if a concise version of the explanation they gave in their cover letter for the sufficiency of the limited sampling of the four production systems is included in the discussion.  I think that such an explanation will alert the readers to the limitations of the study and will improve future such studies by encouraging the sampling of more than one site for each system.   I also think that variability of individual plots within each production system should be included.  This information will enable readers of the paper to assess the confidence in the comparison of the differences between the four production systems.  I recognize that this study involved substantial effort and that there is new information generated hat can be used as a baseline for future studies so it should be published so long as its limitations are made clear.

Reviewer 2 Report

Authors have addressed the majority of the comments and consequently their paper can be accepted for publication 

Adequate, only minor editing
